# Meshless Electrophysiological Modeling of Cardiac Resynchronization Therapy—Benchmark Analysis with Finite-Element Methods in Experimental Data

Carlos Albors [1,*], Èric Lluch [2], Juan Francisco Gomez [3], Nicolas Cedilnik [4], Konstantinos A. Mountris [5,6], Tommaso Mansi [7], Svyatoslav Khamzin [8], Arsenii Dokuchaev [8], Olga Solovyova [8], Esther Pueyo [5,6], Maxime Sermesant [4], Rafael Sebastian [9], Hernán G. Morales [10] and Oscar Camara [1,*]

1   Sensing in Physiology and Biomedicine (PhySense), Department of Information and Communication Technologies, Universitat Pompeu Fabra, 08018 Barcelona, Spain
2   Digital Technology and Innovation, Siemens Healthineers, 91052 Erlangen, Germany; eric.lluch@gmail.com
3   Valencian International University, 46002 Valencia, Spain; juanfrancisco.gomez@campusviu.es
4   INRIA, Université Côte d'Azur, Epione Team, 06902 Sophia Antipolis, France; nicoco@nicoco.fr (N.C.); maxime.sermesant@inria.fr (M.S.)
5   Aragón Institute of Engineering Research, IIS Aragón, University of Zaragoza, 50018 Zaragoza, Spain; konstantinos.mountris@gmail.com (K.A.M.); epueyo@unizar.es (E.P.)
6   CIBER in Bioengineering, Biomaterials & Nanomedicine (CIBER-BBN), 50018 Zaragoza, Spain
7   Digital Technology and Innovation, Siemens Healthineers, Princeton, NJ 08540, USA; tmansi@its.jnj.com
8   Institute of Immunology and Physiology, Ural Branch of the Russian Academy of Sciences (UB RAS), 620049 Yekaterinburg, Russia; svyatoslav.khamzin@gmail.com (S.K.); zodelheim@gmail.com (A.D.); o.solovyova@iip.uran.ru (O.S.)
9   Computational Multiscale Simulation Lab (CoMMLab), Universitat de València, 46100 Burjassot, Spain; rafael.sebastian@uv.es
10  Dassault Systèmes, 78140 Vélizy-Villacoublay, France; hmorales81@gmail.com
*   Correspondence: carlos.albors@upf.edu (C.A.); oscar.camara@upf.edu (O.C.)

**Abstract:** Computational models of cardiac electrophysiology are promising tools for reducing the rates of non-response patients suitable for cardiac resynchronization therapy (CRT) by optimizing electrode placement. The majority of computational models in the literature are mesh-based, primarily using the finite element method (FEM). The generation of patient-specific cardiac meshes has traditionally been a tedious task requiring manual intervention and hindering the modeling of a large number of cases. Meshless models can be a valid alternative due to their mesh quality independence. The organization of challenges such as the CRT-EPiggy19, providing unique experimental data as open access, enables benchmarking analysis of different cardiac computational modeling solutions with quantitative metrics. We present a benchmark analysis of a meshless-based method with finite-element methods for the prediction of cardiac electrical patterns in CRT, based on a subset of the CRT-EPiggy19 dataset. A data assimilation strategy was designed to personalize the most relevant parameters of the electrophysiological simulations and identify the optimal CRT lead configuration. The simulation results obtained with the meshless model were equivalent to FEM, with the most relevant aspect for accurate CRT predictions being the parameter personalization strategy (e.g., regional conduction velocity distribution, including the Purkinje system and CRT lead distribution).

**Keywords:** electrophysiology; parameter optimisation; smoothed particle hydrodynamics; meshless model; cardiac resynchronization therapy; CRT-EPiggy19 challenge

## 1. Introduction

Cardiovascular diseases (CVDs) are one among the leading causes of death worldwide, accounting for 32% of all global deaths [1,2]. The high prevalence of CVD leads to substantial health and economic expenses, as it is one of the most critical challenges in healthcare. Heart failure (HF) is a cardiac pathology that causes CVDs; a non-negligible

number of HF patients have left ventricle (LV) heart's dyssynchrony [3] induced by a left bundle branch block (LBBB) [4,5]. LBBB patients exhibit an abnormal His–Purkinje system, which produces a delay of activation between the interventricular septum and LV-free wall [6].

Cardiac resynchronization therapy (CRT) has demonstrated in randomized clinical trials to be an effective treatment for patients having (i) symptomatic HF; (ii) depressed left ventricular ejection fraction (EF < 35%); (iii) evidence of ventricular dyssynchrony by a prolonged QRS complex (>120 ms). CRT enhances cardiac structure and function through reverse remodeling [4,7,8]. The most consolidated methodology to deliver CRT, biventricular pacing (BiV-CRT), creates an artificial pacemaker in both ventricles and right atrium to resynchronize the electrical activation and, thus, the mechanical contraction between the LV septal and lateral walls at every cardiac beat [9,10]. Nevertheless, more than 30% of patients fulfilling the criteria for CRT implantation do not respond to the therapy (non-responders, NR), although ratios differ according to the applied definition and criteria [11,12]. One of the main reasons for the high rate of CRT non-responders is the use of too simple indices for patient selection (e.g., EF, QRS, New York Heart Association class). Beyond optimization of patient selection, the correct electrode placement is a key factor to reduce the number of CRT-negative responses. Potential therapeutic alternatives to traditional BiV-CRT are emerging based on optimization of lead placement and number [13] or on new physiological stimulation modalities [14].

Computational electrophysiological models can be valuable tools for a better understanding of pacing-based therapies such as CRT, providing additional information to physicians and device manufacturers to improve therapy efficacy. The interested reader is referred to Niederer et al. [15] and Lee et al. [16] for comprehensive reviews on computational models in cardiology and specific to LBBB and CRT, respectively. More recently, some studies have focused on CRT response optimization through electromechanical models including coronary perfusion [17], or myocardial strains with a complete cardiovascular system, adding both atria as well as systemic and pulmonary circulations [18]. Other studies particularly investigate lead placement. For instance, Albatat et al. [19] analyzed the benefits of multi-site pacing in CRT patients with myocardial infarction. Carpio et al. [20] explored RV lead optimization in a complete simulated torso, while Oomen et al. [21] used fast electro-mechanical simulations to study the role of post-infarction ischemia in reverse LV remodelling following CRT.

Patient-specific personalization plays an important role to make computational models more realistic. However, detailed electrical and mechanical information of the heart is needed, often only available from invasive techniques [16]. Due to the difficulties of obtaining the required in vivo data in humans at different stages of the disease (e.g., from healthy to LBBB and with a CRT device), the validation of CRT computational models is challenging.

Cardiac computational modeling can be improved by translating pre-clinical data into patient-specific models, linking animal and clinical research. For example, as a result of the participation in the Cardiac Electrophysiological Simulation Challenge (CESC'10) MICCAI-STACOM workshop (https://stacom.github.io (accessed on 26 April 2022)), several research groups [22] developed a pipeline integrating different modeling approaches to predict depolarization isochrones from optical mapping data of a perfused ex vivo porcine heart with different pacing conditions [23], acquired at the Sunnybrook Health Sciences Centre, Toronto, Canada. However, experimental data were available for two cases.

Some years later, Rigol et al. [24] developed a swine model of LBBB to study the link between electrical and mechanical dyssynchrony, and their correction with CRT. The authors generated a unique dataset with signal, multi-modal images and electro-anatomical maps at different stages of the disease in tens of infarcted and non-infarcted animals. Soto Iglesias et al. [25] proposed advanced visualization techniques and metrics to quantify the differences in electrical activation patterns at baseline, LBBB and CRT stages. A subset of the database was the foundation for the organization of the CRT-EPiggy19 challenge (https://crt-epiggy19.surge.sh/ (accessed on 15 April 2022)) at the MICCAI-STACOM19 workshop,



which is available open access in a public repository (https://zenodo.org/record/3249511 (accessed on 18 April 2022)). More recently, Ramirez et al. [26] also developed a swine model of the heart that was coupled with electrophysiological models to study advanced biomaterial injection therapies for ischemic heart failure.

Participants at the CRT-EPiggy19 challenge adopted different modeling approaches to predict the electrical activation after CRT. Khamzin et al. [27] and Cedilnik and Sermesant [28] developed personalization strategies based on genetic algorithms to estimate regional conduction velocities with simple but fast phenomenological Eikonal-based models. Meanwhile, Gomez and Sebastian [29] used a more detailed Ten Tusscher–Panfilov [30] for cellular electrophysiology, considering transmural heterogeneity and electrical propagation by a monodomain model. After the challenge, other researchers have used the provided data to better understand cardiac physiology and pacing-based therapies [31].

All the aforementioned approaches are based on solving the electrophysiological model equations with the finite-element method (FEM) as a numerical technique based on a mesh discretization of the biventricular heart geometry, as it is the common choice in cardiac modeling [32]. In FEM, the computational domain is divided into discrete subsets of interconnected nodes as elements. However, the explicit connectivity required in the domain leads to great difficulty in generating the irregular patient-specific cardiac meshes, which then becomes a tedious, manual, highly interactive, and time-consuming process. Moreover, the reliability of the simulation results is highly dependent on the quality of the built geometrical mesh [33]. Additionally, mesh distortion that can occur during large cardiac deformations enforces the use of remeshing algorithms to restore mesh shape and numerical accuracy, thereby increasing the computational cost and efforts [34]. Meshless methods are an interesting alternative to avoid meshing difficulties, since the spatial domain is composed of an unstructured particle cloud without connectivity. Therefore, the meshless domain construction procedure can be used for any type of complex geometry. In addition, large deformations or the linking of meshes with different spatial resolution, often necessary in cardiac electromechanics, can be better handled with meshless methods than with FEM, as FEM-based connectivity does not need to be satisfied. For instance, authors in [35] have shown the potential of meshless methods for fluid–structure interaction (FSI) applications, which are extremely time-consuming for mesh-based methods.

Meshless approaches have already been applied to cardiac modeling. For example, Wong et al. [36] used an element-free Galerkin meshless method for modeling cardiac mechanics. On the other hand, Lluch et al. [37] developed meshless methods based on smoothed particle hydrodynamics (SPH) meshless technique for modeling cardiac mechanics. The same authors later [38] employed genetic algorithms to calibrate a SPH-based fully coupled electro-mechanical model of the heart with high-resolution imaging and invasive in vivo measurements from a healthy canine heart. Recently, Mountris and Pueyo [39] proposed a meshfree mixed collocation method with interpolating trial functions to solve the monodomain equations for cardiac electrophysiology and the O'Hara ventricular cell model [40], which was applied to one of the CRT-EPiggy19 challenge dataset under healthy and LBBB conditions.

In this manuscript, we present a benchmark analysis of a meshless SPH method with finite-element methods for the prediction of cardiac electrical patterns in CRT, based on a subset of the CRT-EPiggy19 dataset, including infarcted and non-infarcted cases. A data assimilation strategy was designed to personalize the most relevant parameters of the electrophysiological simulations and identify the optimal CRT lead configuration.

## 2. Materials and Methods

### 2.1. CRT-EPiggy19 Data and Experiments

The experiments to create the CRT-EPiggy19 data were performed at Hospital Clínic de Barcelona, Spain, after animal handling approval of the Institutional Review Board and Ethics Commitee of the hospital. In the animals, radiofrequency ablations were carried out to induce LBBB, where half of them presented a myocardial infarction located at the septal wall with different levels of transmurality. Then, a CRT device was implanted to

later study the effects of the therapy. More details of the experimental protocol can be found in Rigol et al. [24].

A subset of the CRT-EPiggy19 data was used in our study, including three cases for training and testing, respectively (two non-infarcted and one infarcted dataset in each group). In the dataset provided by the challenge organizers, image segmentation and biventricular finite-element mesh reconstruction were performed using an in-house Siemens algorithm applied on cine sequences of Magnetic Resonance Imaging (MRI), acquired from the swines during the experimental studies. The scar was manually segmented and quantified from delay-enhancement MR images. In terms of electrophysiological data, anatomical point-based reconstructions from CARTO XP of epicardial and endocardial layers were obtained at baseline, LBBB and CRT phases. The electro-anatomical map (EAM) clouds of points were then interpolated onto the MRI biventricular FEM meshes through a quasi-conformal mapping method [25]. Finally, a rule-based method [41] was used for the generation of the cardiomyocyte orientation in each mesh. In addition, regional labels (AHA regions, ventricle definition, endo- and epi-cardial wall distinction) and scarred AHA segments were also included in the models. In the training set, each porcine model was reported in two distinct pathologic stages: with a block in the left bundle branch of the purkinje system and after CRT. For the testing dataset, only the LBBB stage was provided to personalize the electrophysiological models and used them to predict CRT electrical patterns. The RV endocardium was not acquired in the EAM data; therefore, the analysis was centered on the endocardial LV layer and biventricular epicardial layer.

### 2.2. Meshless Model Based on Smoothed Particle Hydrodynamics

The total Lagrangian meshless method (TL-SPH) developed by Lluch et al. [38] was used in our experiments. As a meshless model, SPH is easy to parallelize, and memory efficient. Additionally, it is mathematically rigorous since it satisfies the Kronecker's delta property. Figure 1 illustrates the developed meshless SPH-based modeling pipeline to predict CRT electrical patterns in the experimental data. The first step of the pipeline consisted on discretizing the continuous domain provided by the biventricular meshes of the porcine hearts with a cloud of particles without connectivity, where each particle had the following individual properties: three-dimensional position, cardiomyocyte orientation, tissue type, initial impulse, conduction velocity, area and volume.

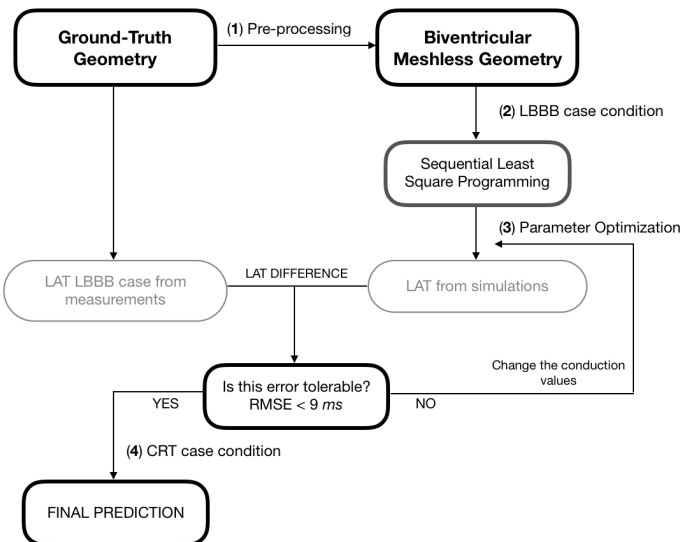

**Figure 1.** Scheme of the developed meshless modeling pipeline to predict the electrical patterns in experimental data after cardiac resynchronization therapy. LBBB: left bundle branch block. RMSE: root mean square error. LAT: local activation time.

To determine the particle properties a Gaussian Kernel function was enforced, defining the number of neighboring particles around each particle of interest that is then employed

to estimate the area and volume indices. Parameters such as the kernel size (i.e., smoothness length of the Gaussian kernel) and the geometry discretization according to the number of particles were key factors to determine when setting up the simulation. To define the optimal values of the Gaussian kernel function, sensitivity analyses of particle resolution and kernel size were performed. Several simulations were carried out fixing the kernel size and increasing the particle numbers; configurations with $15 \times 10^3$ (2 h simulation), $20 \times 10^3$, $80 \times 10^3$ and $100 \times 10^3$ (11 h simulation) particles were studied in one of the analyzed geometries. The kernel size was inversely proportional to the number of particles to avoid excessive computational cost; kernels from 3.5 to 9 mm were tested in intervals of 0.5 mm. A kernel size of 6.5–8.5 mm was finally defined, as a function of the swine model morphology and required conduction velocities (i.e., larger kernels for higher conduction velocities and morphologies with higher curvature), in combination with geometries of $15 \times 10^3$ particles. As shown in [42], configurations with higher number of particles (e.g., $50 \times 10^3$) and smaller kernel sizes (e.g., 3 mm), computational costs exponentially increases without a substantial accuracy gain, which will hamper the parameter optimization process. Furthermore, we also analyzed the effect of the time-step, testing values of $10^{-3}$, $10^{-4}$, and $10^{-5}$ in one of the studied cases for LBBB simulations. The computational cost associated with each time-step value was of >42 min, around 20 min and around 7 min, providing RMSE of 6.2 ms, 6.8 ms, and 7.9 ms, respectively. A time-step value of $10^{-4}$ was finally chosen as a trade-off between computational cost and result accuracy. The total simulated time was of 0.15 s, based on the total activation times of the available EAM dataset (i.e., most cases with TAT < 0.1 s).

### 2.3. Electrophysiological Model

The simplified reaction–diffusion Mitchell–Schaeffer (MS) electrophysiological model [43], together with a diffusion term [42], was used at the cellular level. The MS method allowed us to simulate the electrical activation sequence of the swine hearts with an ionic model of the ventricular action potential duration (APD) composed of only two currents: one inward and one outward. The computation of the voltage and depolarization phase over time is performed with the following partial differential equations:

$$
\begin{cases}
\dfrac{\partial v}{\partial t} = div(\boldsymbol{D}\nabla v) + \dfrac{wv(1-v)}{\tau_{in}} - \dfrac{v}{\tau_{out}} + \boldsymbol{I}_{app} \\[2mm]
\dfrac{\partial w}{\partial t} = \begin{cases} \dfrac{1-w}{\tau_{open}} \text{if} vs. < v_{gate} \\[2mm] \dfrac{-w}{\tau_{close}} \text{if} vs. > v_{gate} \end{cases}
\end{cases} , \tag{1}
$$

where $I_app \in \mathbb{R}$ describes the initial stimulus of the transmembrane potential $v \in \mathbb{R}$, $w \in \mathbb{R}$ controls the depolarization phase, and $v_{gate} \in \mathbb{R}$ determines where the APD starts. Furthermore, $\tau_{open}$, $\tau_{close}$, $\tau_{in}$, and $\tau_{out} \in \mathbb{R}$ govern the duration of the four stages of the APD (i.e., initiation, plateau, decay, and recovery). The diffusion term, $div(\boldsymbol{D}\nabla v)$, includes cardiomyocyte orientation, with the diffusion tensor, $\boldsymbol{D} \in \mathbb{R}^{3\times3}$, defined as in [44]:

$$
\boldsymbol{D} = (f \otimes f(1 - ar) + I \cdot ar) \cdot d \tag{2}
$$

There are three main parameters in Equation (2) to take into account: the cardiomyocyte orientation vector, $f \in \mathbb{R}^3$; the diffusion coefficient, $d \in \mathbb{R}$, which controls the action potential propagation speed; and the anisotropic ratio, $ar \in \mathbb{R}$, which determines the relation between conduction velocities and cardiomyocyte orientation (e.g., $ar = 1$ will define an isotropic behavior). We tested different values for $ar$ (from 0.01 to 0.5), finally fixing to 0.01 (i.e., giving more weight to cardiomyocyte orientation) for all cases. The cardiomyocyte orientation was provided in all studied biventricular meshes by the CRT-Epiggy19 organizers from the rule-based model proposed by Doste et al. [41], which is adapted to replicate histological data of both left and right ventricles. Finally, $\boldsymbol{I} \in \mathbb{R}^{3\times3}$ defines the

identity matrix and $\otimes$ the tensor product. Overall, only six parameters are necessary in the MS model, which is convenient for model personalization.

### 2.4. Left Bundle Branch Block Simulation with Personalized Parameters

An initial stimulus was set in the atrio-ventricular (AV) node, with an average of 60 particles, being identified from the earliest activated points in the EAM data of each case, to initiate the simulated electrical pattern over the two ventricles. The Purkinje (PK) system, which has fast conduction velocities, needs to be incorporated in the model for simulating a LBBB and disrupt the normal electrical propagation in the LV branch. Therefore, particles located in the lower (i.e., closer to the apex) half of the endocardial RV (around 500 particles) and the lower third of the LV (around 300 particles), if no scar was present, were labeled as Purkinje (see Figure 2), following the distribution of PK–myocardial junctions found in PK-based simulation studies [45].

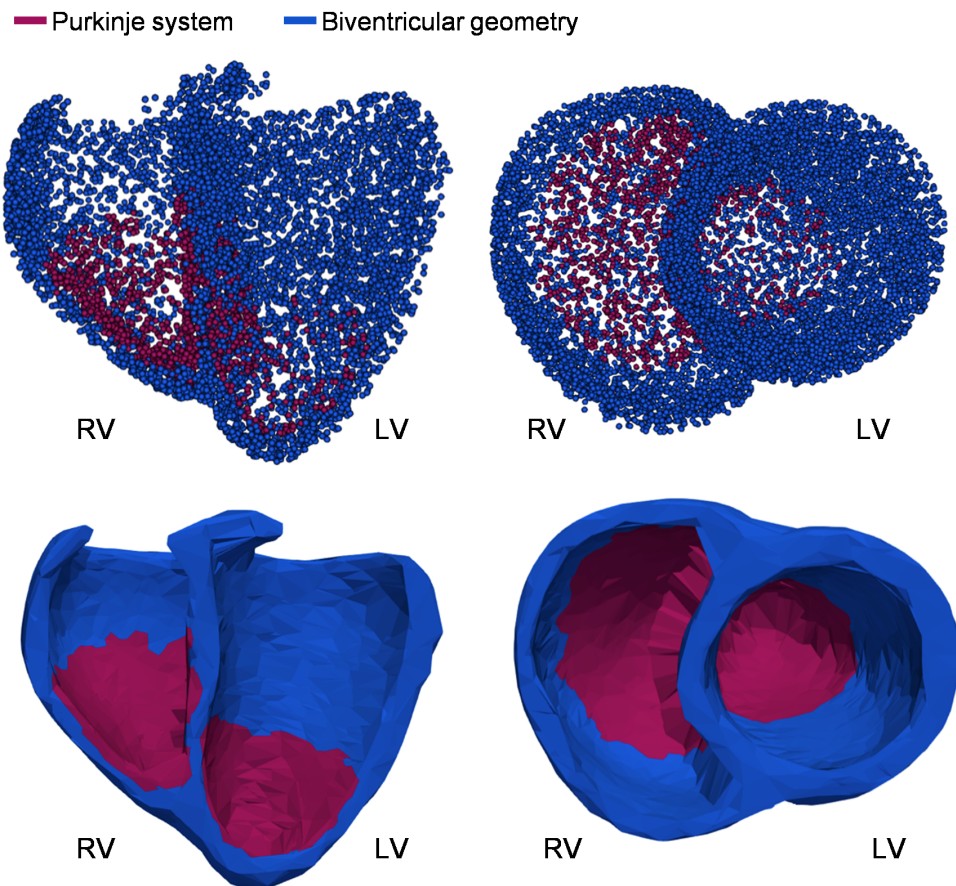

**Figure 2.** Biventricular geometries with particles labelled as regular myocardial tissue (in blue) and Purkinje system (in red), used for the meshless solver. Top: point cloud representation. Bottom: mesh-based triangulation from applying the Delaunay algorithm to the cloud of points, for visualization purposes.

Including PK particles in the LV, could seem contradictory for simulating LBBB electrical patterns. However, it was necessary to consider the LV retrograde activation due to the transmurality of the PK system in pigs [46], leading to latest activation points being located at the basal LV in several cases.

As other participants at the CRT-EPiggy19 challenge [28,29], we also added a second impulse in the electrophysiological simulations to replicate an early activation of the RV epicardium observed in the EAM data (see $RV_{epi}$ initial activation point in Figure 3). Besides the expected activation induced by the AV node, the septomarginal trabecula may have a role in the fast activation of the RV that needs to be incorporated to replicate the

electrophysiological measurements. The dynamics of the electrical pattern in LBBB cases are displayed in Figure 3, starting from the two stimulus (1 in the Figure 3), followed by the propagation to the biventricular apex (2 in the Figure 3), and the propagation from the LV apex to the base (3 in the Figure 3), with fast activation in the endocardium and slow in the epicardium.

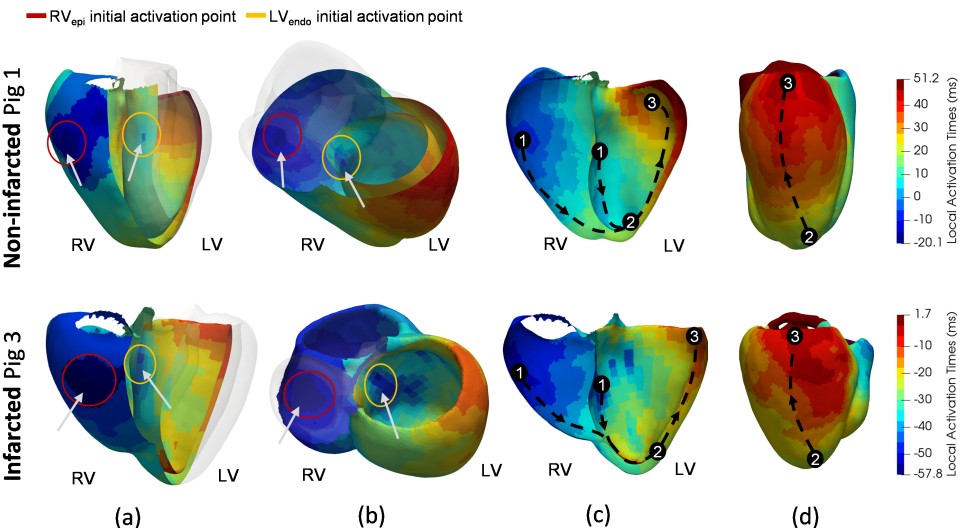

**Figure 3.** Electroanatomical maps of an infarcted and a non-infarcted training case in left bundle branch block condition. In (**a**,**b**), both initial activation points (1) in the RV epicardial layer (red circle) and LV endocardial one (yellow circle) are shown. At (**c**,**d**), the numbering sequences describe the followed electrical pattern to fully activate the biventricular geometries: starting from the initial stimulus (1), following to the biventricular apex (2), and propagating to the left ventricular base (3). The colourscale represents the local activation times, from earliest to the latest activation points, in blue and red, respectively. $RV_{epi}$: right ventricle epicardium. $LV_{endo}$: left ventricle endocardium.

The local conduction velocity (CV) values defined in each geometry, guiding the wave propagation speed in the direction established by the modeled cardiomyocyte orientation, was one of the main parameters affecting the simulated electrical pattern. However, it is not simple to set up the number of heterogeneous conductivity regions: different values at each voxel would both be impractical (too many parameters to optimise) and does not make sense in relation with the sparsity of the available electroanatomical data; too few regions would not consider the existing CV heterogeneity (e.g., faster CV in PK system, complex electrical propagation in the septum due to discontinuities in cardiomyocyte orientation [41], presence of scar, etc.). Consequently, we performed a sensitivity analysis to determine the optimal number of different regions with local conductivities to optimize, from only a single region considering the whole biventricular geometry, to 21 regions including the 17 AHA segments. In total, the following seven regional CV configurations were tested:

- 1 region (*whole biventricular geometry*).
- 2 regions ($RV - LV$).
- 3 regions ($RV - Purkinje\ system - LV$).
- 4 regions ($RV_{epi} - RV_{endo} - LV_{epi} - LV_{endo}$).
- 5 regions ($RV_{epi} - RV_{endo} - Purkinje\ system - LV_{epi} - LV_{endo}$).
- 6 regions ($RV_{epi} - RV_{endo} - Purkinje\ system - Septum - LV_{epi} - LV_{endo}$).
- 21 regions ($RV_{epi} - RV_{endo} - Purkinje\ system - 17\ LV_{AHA_{segments}} - LV_{endo}$).

The optimization of the CV distribution in each analyzed case was performed with the constrained non-linear Sequential Least Squares Programming algorithm. The cost function was based on minimizing the root mean square error of each particle activation time

between simulation results and the EAM-based electrical patterns. An iterative method was used for parameter optimization, updating the five regional CVs until the best possible fit was obtained. The choice of a constrained algorithm was made so that: (1) conductivity values always were positive; (2) a purkinje system always being the fastest regional layer; (3) the lowest conductivity value always was in the necrotic/scar zone for infarcted cases. In the end, an average of 70 simulations were performed for each analyzed case, mainly for the optimization of the CV configuration.

### 2.5. Simulation of Cardiac Resynchronization Therapy

Once model parameters were personalized with the SLSQP optimization algorithm to better replicate the electrical pattern of the LBBB data, the next step was to simulate CRT using the same personalized parameters (see Figure 1). Additional initial stimulus were incorporated in the model, simulating the LV and RV leads of CRT. The position of the CRT leads in the training cases was determined by identifying the earliest activated points in the provided electroanatomical maps. In the testing cases, as EAM data were not available, several lead configurations were evaluated to find the one furnishing better evaluation metrics, as described below.

### 2.6. Evaluation Metrics and Experiments

As mentioned above, the root mean square error difference between the local activation time (e.g., time when each particle activates, with the initial stimulus as reference) given by the simulations and the EAM measurements, integrated over each particle of the biventricular geometries, was used in the parameter optimization in the training cases. As for testing, global and regional metrics were used to evaluate the prediction accuracy of different modeling strategies in each analyzed case.

First, the total activation time (TAT) required to activate the whole biventricular geometry from the initial impulses, was employed as a general metric. Additionally, as proposed by Soto Iglesias et al. [25], we computed some activation delays to better characterize regional patterns, specifically, the inter-ventricular delay (IVD), which is time difference between earliest activation points of both ventricles (LV and RV) in the epicardial layer; and the left ventricular transmural delay (LV-TD), defined as time difference between LV layers (epi- and endocardium) first activated points. Finally, we also estimated the recovery as follows:

$$Recovery = \frac{TAT_{baseline} - TAT_{LBBB}}{TAT_{LBBB} - TAT_{CRT}} * 100, \tag{3}$$

which indicates the percentage of how close the TAT is to the baseline after applying CRT. Finally, we created histograms of the percentage of activated tissue over time for the right and left epicardial regions, which provides an intuitive visualization of the different intra- and inter-ventricular delay differences between LBBB and CRT conditions (including distinct lead configurations).

After the sensitivity analyses of different modeling choices of the SPH-based solution (e.g., number of particles, kernel size, time step), as explained in Section 2.2, the initial experiments in our study consisted on personalizing model parameters (e.g., regional conduction velocities) with the EAM data in the three studied training cases in LBBB condition. Subsequently, the resulting regional conduction velocity distribution was used for modeling CRT, using the lead position provided by the challenge organizers in the training dataset. The initial stimulus characteristics (e.g., location, depolarization times) were maintained in all models).

The meshless simulation results were qualitatively and quantitatively compared with the ones provided by an FEM-based method [29] presented in the CRT-EPiggy19 challenge, since it was the only participant processing the three analyzed training cases. Additionally, metric comparisons are already made with the mesh-based methodology presented by another challenge participant [27], which reduced the biventricular geometry mesh res-

olution to $12 \times 10^3$ elements to facilitate the exploration of a larger range of parameters (>30 different conduction velocity regions) in an optimization process using the L-BFGS optimization algorithm.

In the CRT-EPiggy19 challenge, EAM data of testing cases after CRT were not provided, thus lead location was unknown. For this reason, we tested four different lead locations (Figure 4) in each testing geometry to determine the one providing the best recovery: (1) RV apex and LV basal region ($RV_{apx} - LV_{bas}$); (2) RV and LV Apex ($RV - LV_{apx}$); (3) earliest and latest point activation from the LBBB cases in the EAM ($Early - Late_L AT$); (4) RV Outflow track (RVOT) septal and LV basal region ($RVOT_{sep} - LV_{bas}$). If recovery was similar in different lead configurations, TAT and delay values were analyzed to choose the final lead configuration.

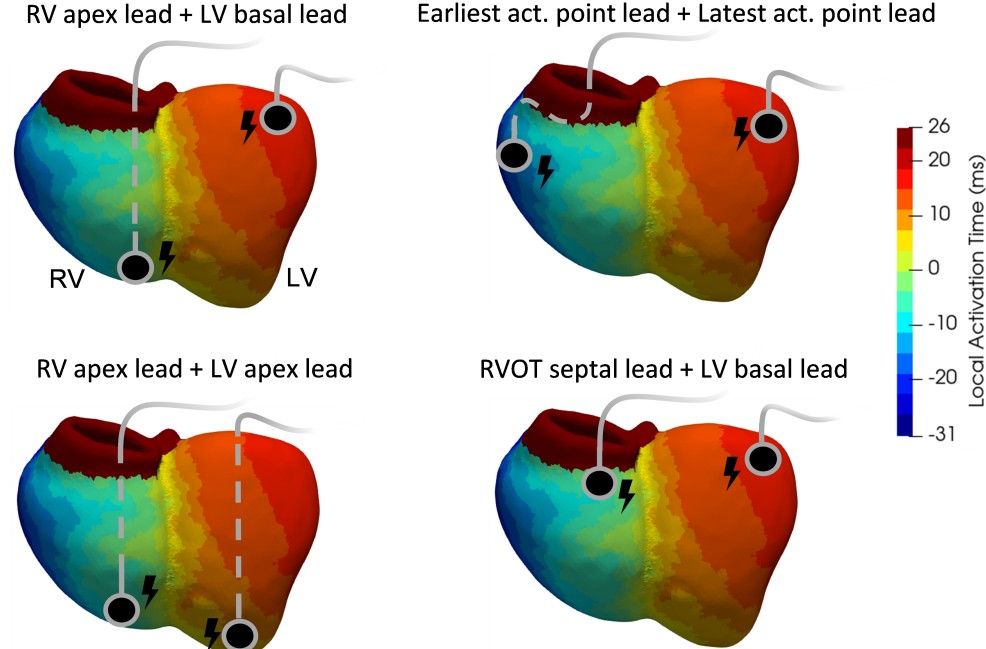

**Figure 4.** Different configuration of cardiac resynchronization leads analysed in the testing cases. LV/RV: left and right ventricle, respectively. Earliest/Latest act: Earliest/Latest activation. RVOT: right ventricular outflow tract.

For comparison purposes, colourmaps representing the electrical activation patterns of the figures have been adjusted by setting the initial depolarization of the RV of each model (local activation values) as the initial times and dividing them into several isochrones. For visualization, the Open Source Paraview (ParaView, v.5.8) (https://www.paraview.org (accessed on 1 April 2022)) software tool was used. Computational resources for the meshless electrophysiological models consisted of a Nvidia RTX 2080 Ti GPU and an i9-9900k CPU executed in Code::Blocks software (Code::Blocks IDE, v.16.01).

## 3. Results

### 3.1. Training Data

The sensitivity analysis to determine the best regional distribution of conduction velocities in the LBBB condition resulted in best fittings of simulations with EAM data when increasing the number of regions, with a RMSE of 6.4 ms and 5.3 ms in the non-ischemic and ischemic models, respectively, for 21 regions (vs. 6.7 ms and 5.8 ms in the non-ischemic and ischemic models, respectively, for 5 regions). However, when applied to CRT data, electrophysiological simulations with 6 and 21 regions produced larger errors than with 5: 10.2 ms and 9.8 ms in the non-ischemic and ischemic cases, respectively, for 21 regions, and 9.3 ms and 7.7 ms in the non-ischemic and ischemic cases, respectively, for 5 regions. Therefore, five regions were finally chosen for the conduction velocity distribution in the

remaining simulations. The optimization process took between 10 and 25 h to converge (15–25 min per simulation), depending on the studied case.

### 3.1.1. Left Bundle Branch Block Simulations

Table 1 summarizes the accuracy obtained with the meshless SPH-based in the training dataset, as quantified by the metrics detailed above. Equivalent results were obtained for both LBBB and CRT conditions. Figure 5 shows the local activation time maps for a non-infarcted and an infarcted case of the training database at LBBB and CRT conditions, provided by the EAM, and from FEM- and SPH-based simulations.

**Table 1.** Metrics characterizing the electrical activation maps in training cases from measurements and meshless simulations. EAM: electroanatomical maps. SPH-Sim: Simulation with smoothed particle hydrodynamics meshless method. LBBB: left bundle branch block. CRT: cardiac resynchronization therapy. TAT: total activation time. LAT-RMSE: local activation time root mean square error. IVD: inter-ventricular delay. LV-TD: left ventricle transmural delay. (*) indicates an infarcted pig.

| | Pig 1 | | | | Pig 2 | | | | Pig 3 (*) | | | |
| | EAM | | SPH-Sim | | EAM | | SPH-Sim | | EAM | | SPH-Sim | |
| | LBBB | CRT | LBBB | CRT | LBBB | CRT | LBBB | CRT | LBBB | CRT | LBBB | CRT |
|---|---|---|---|---|---|---|---|---|---|---|---|---|
| TAT (ms) | 72.0 | 70.0 | 70.0 | 66.0 | 66.0 | 45.0 | 78.0 | 39.0 | 59.0 | 35.0 | 49.7 | 46.0 |
| LAT RMSE (ms) | | | 6.8 | 7.9 | | | 9.4 | 7.7 | | | 5.1 | 6.6 |
| 17 IVD (ms) | 18.0 | 7.3 | 18.6 | 11.4 | 19.8 | −3.3 | 14.3 | 0.0 | 17.7 | −12.5 | 16.9 | −4.8 |
| LV-TD (ms) | 7.2 | 0.0 | 9.0 | 2.0 | 9.9 | −6.6 | 13.0 | 0.4 | 11.8 | −5.2 | 18.4 | 0.6 |
| Recovery (%) | −2.6 | | −8.0 | | 47.7 | | 69.6 | | 342.9 | | 185.0 | |

The initial impulses set for the non-ischemic case shown in Figure 5 (Pig 1) were established with a difference of 12 ms between them. The first, located in the RV endocardium, was set at 2 ms, and the second, the septal one, at 14 ms. Table A1 and Figure 5 revealed similar activation patterns in the biventricular epicardium with an average regional LAT RMSE of 5.7 ms compared to the EAM data. However, larger differences were observed in the LV endocardial layer, increasing the regional error to 9.1 ms (Table A1). The remaining metrics (e.g., TAT, IVD and LV-TD) were similar between meshless simulations and measurements, with a difference <3 ms, while the overall error was 6.8 ms, as reported in Table 1.

For the infarcted case shown in Figure 5 (Pig 3), the initial stimulus was placed as in Pig 1, but at 2.5 ms in the RV endocardial layer, and in the septal area at 12 ms to better match the EAM data. Figure A3 in the Appendix shows simulation results obtained with only one stimulus. The scar in Pig 3, which had a transmurality of 86%, was located in the septo-apical and antero-septal LV regions. In the scar region, a 75% conduction velocity reduction with respect to the Purkinje system was established from the best simulation result. Although the electrical activation pattern provided by the SPH simulations was very close to the EAM measurements (5.1 ms average LAT error), differences in conductivity were appreciated between both ventricles. The posterior basal region of the RV epicardial layer had a slower activation than the ground-truth data, whereas LV layers (endo- and epicardium) had a higher conductivity. The metrics in Table 1 show differences greater than 6 ms for LV-TD, 9 ms for TAT, and less than 1 ms for IVD.

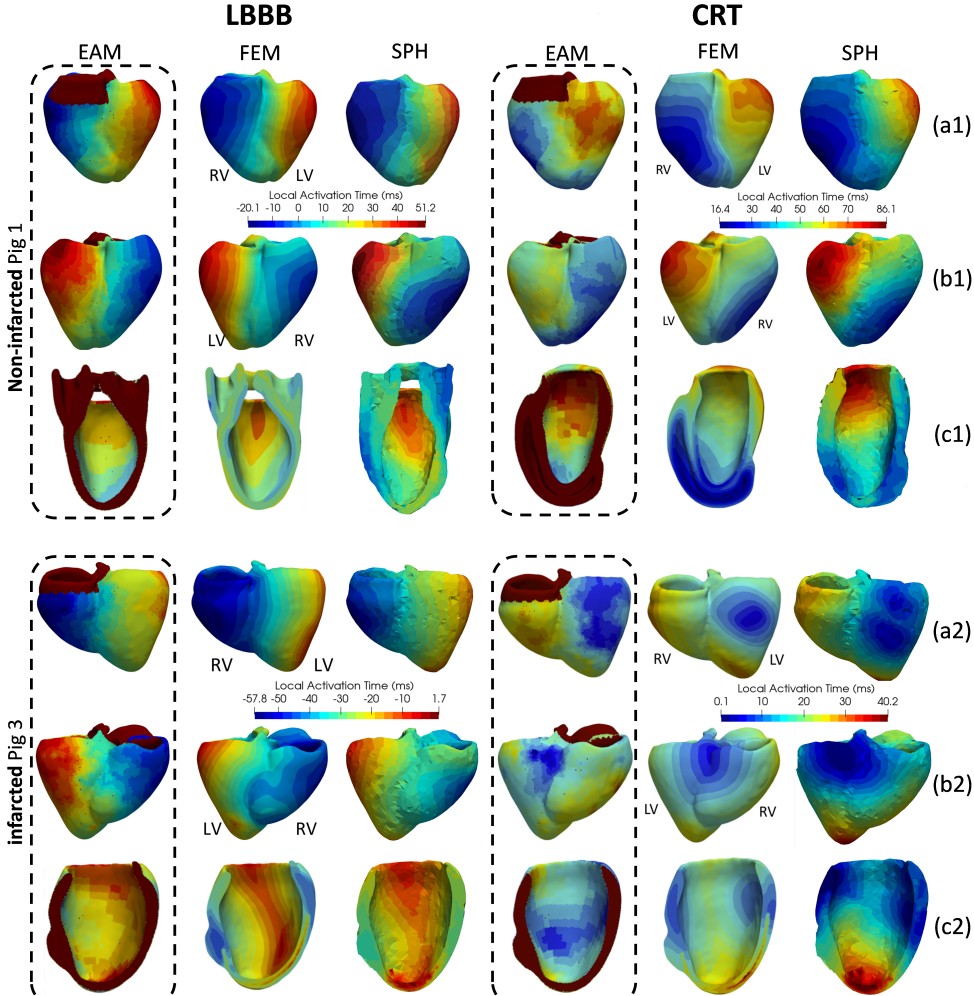

**Figure 5.** Local activation time maps for a non-infarcted and an infarcted (top and bottom panels corresponding to Pig 1 and Pig 3, respectively) case of the training database in left bundle branch block (LBBB) and cardiac resynchronization therapy (CRT) conditions, provided by the electroanatomical measurements (EAM) and electrophysiological simulations performed with a finite-element method (FEM) and a meshless (SPH) model. (**a1,a2**) and (**b1,b2**) correspond to anterior and posterior biventricular epicardial visualizations. (**c1,c2**) show endocardial view of the left ventricle (LV) lateral wall. RV: right ventricle.

Table 2 shows the conduction velocity values estimated by the SPH-based model in the five selected regions for an ischemic and a non-ischemic cases of the LBBB training dataset. Additionally, the corresponding parameters obtained with the FEM-based approach of Gomez and Sebastian [29] on the same cases are also included for comparison purposes. The reader can be referred to Figure A2 for a visual representation.

**Table 2.** Conduction velocity values (*m/s*) estimated by the SPH- and FEM-based solvers [29] for an ischemic and non-ischemic training cases at LBBB scenario. SPH: Smoothed particle hydrodynamics meshless method. FEM: Finite element method. RV epi: Right ventricle epicardium. RV endo: Right ventricle endocardium. LV: Left ventricle. PK: Purkinje system.

| | SPH-Based | | | | | | | FEM-Based | |
|---|---|---|---|---|---|---|---|---|---|
| | RV endo | RV epi | LV endo | LV epi | Scar | Average heart tissue | PK | Average heart tissue | PK |
| Ischemic | 1.53 | 1.40 | 1.36 | 1.62 | 0.49 | 1.30 | 1.69 | 1.78 | 1.30 |
| Non-ischemic | 0.83 | 0.65 | 0.63 | 0.51 | - | 0.65 | 2.40 | 0.50 | 2.60 |

### 3.1.2. Cardiac Resynchronization Therapy Simulations

The configuration of the CRT leads was initially positioned close to the apical regions of both ventricles in the non-infarcted training case shown in Figure 5, following the information provided by the organizers of the CRT-EPiggy19. In the EAM data, the mid-apical lead location on the lateral wall of the RV endocardial layer resulted in fast epicardial conduction, specifically in the posterior part. In contrast, LV lead apicality with weak access to the PK system implied slower activation of its endocardial layer than of the epicardial one, with the former presenting the last activation point. The SPH-based simulation produced the largest differences (9.7 ms of LAT error in Table A1) in the RV. As for the LV, an 8.5 ms LAT error was found, since it was not possible to fully capture the conduction velocity change between the endocardium and epicardium. The metrics summarized in Table 1 present differences between simulations and observations of 2 ms for LV-TD, 4 ms in TAT, and 4.1 ms in IVD with an error of 9.2 ms in the overall LAT.

For the infarcted testing case shown in Figure 5 (pig 3), non-physiological conduction velocities above 2 m/s, specifically in the ischemic zone, were required to match the fast electrical patterns observed in the EAM data (35 ms), with both CRT leads located in the LV epicardial layer (anterior and posterior regions). The parameter optimization process in SPH-based simulations did not capture these high conduction velocities due to the physiological constrains, providing slower values and exhibiting large differences with EAM in the apex, reflected in the 11 ms of TAT and in delay metrics over 6–7 ms. However, the overall LAT error was not large (6.6 ms).

### 3.2. Testing Data

#### 3.2.1. Left Bundle Branch Block Simulations

Table 3 summarizes the accuracy obtained with the meshless SPH-based in the testing dataset. The initial impulses were fixed at 0 ms in the endocardial RV layer and at 9 ms in the septal area for Pig 4, one of the non-infarcted testing cases. The SPH-based simulation correctly replicated the conduction velocities of the LBBB EAM at different layers showing a low error of 5.1 ms, and specifically the RV epicardium with a 4 ms regional error (Table A2). Nevertheless, the anterior part of the LV endocardium showed a greater number of variations, corroborated by a regional error above the mean (6.4 ms in Table A2). The LV epicardial sequence was also similar (5.5 ms regional LAT RMSE in Table A2) in measurements and simulations, with the same latest activation point.

**Table 3.** Metrics characterizing the electrical activation maps in testing cases from measurements and meshless simulations, including the best lead configuration in the cardiac resynchronization therapy scenario. EAM: electroanatomical maps. SPH-Sim: Simulation with smoothed particle hydrodynamics meshless method. LBBB: left bundle branch block. CRT: cardiac resynchronization therapy. TAT: total activation time. LAT-RMSE: local activation time root mean square error. IVD: inter-ventricular delay. LV-TD: left ventricle transmural delay. $RV_{apx} - LV_{bas}$: CRT leads in the right ventricular apex and basal left ventricle. (*) indicates an infarcted pig.

| | Pig 4 | | | | Pig 5 | | | | Pig 6 (*) | | | |
| --- | --- | --- | --- | --- | --- | --- | --- | --- | --- | --- | --- | --- |
| | EAM | | SPH-Sim | | EAM | | SPH-Sim | | EAM | | SPH-Sim | |
| | LBBB | CRT | LBBB | CRT ($RV_{apx} - LV_{bas}$) | LBBB | CRT | LBBB | CRT ($RV_{apx} - LV_{bas}$) | LBBB | CRT | LBBB | CRT ($RV_{apx} - LV_{bas}$) |
| TAT (ms) | 61 | 59 | 56 | 36.8 | 92 | 70 | 76 | 55 | 67 | 49 | 73.7 | 47 |
| LAT RMSE (ms) | | | 5.1 | 13.2 | | | 8.2 | 14.6 | | | 5.9 | 10.7 |
| IVD (ms) | 19.61 | −9.04 | 14.7 | 0.5 | 25.58 | 7.83 | 21.5 | 1.2 | 12.75 | 2.63 | 18.3 | 0.8 |
| LV-TD (ms) | 16.55 | −12.95 | 5.8 | −0.7 | 32.68 | −8.45 | 9.8 | -0.3 | 14.22 | −1.97 | 6.8 | 0.6 |
| Recovery (%) | 17.95 | | 100.84 | | 46.09 | | 68.2 | | 637 | | 666 | |

The EAM of the infarcted testing case (Pig 6) had an initial impulse at the RV endocardium lateral wall, inducing a rapid RV activation, while the LV one was much more gradual. The scar in Pig 6 was located in over the whole septum, with 57% of transmurality. In the SPH-based simulations, the initial stimulus were placed at similar regions of the non-infarcted case but at 2.6 ms and 14 ms for the RV endocardial layer and septal area, respectively. To faithfully represent the ischemic region in the simulations, a reduction of over 87% in the conduction velocity with respect to the PK system was determined by the SLSQP optimization algorithm. The epicardial layer depicted the highest LAT regional error (see Table A2), specifically in the posterior part for the RV and in the LV anterior part. With a LAT-RMSE of 5.9 ms for Pig 6 (Table 3), the LV endocardial layer showed the best fitting (5.7 ms regional LAT-RMSE) for a delayed basal activation of the LV (Table A2).

### 3.2.2. Cardiac Resynchronization Therapy Simulations

In the three analyzed testing cases, the optimal configuration consisted in leads located in the RV apex and the basal LV ($RV_{apx} - LV_{bas}$ configuration), providing the best recovery metric values and overall cardiac resynchronization. The CRT leads were activated at practically the same time in the three testing cases, with a time interval under 0.5 ms between them. The RV was typically triggered prior to the LV (see histograms in Figure 6 for the infarcted testing case), where it was always located the last activation point.

Figure 6 shows the local activation time maps for a an infarcted case (Pig 6) of the testing database in CRT condition, provided by the EAM, and for different simulated lead configurations. Furthermore, histograms of the percentage of electrically activated heart tissue for the right and left epicardial layers are displayed to better represent inter-ventricular delays with different lead configurations. It can be easily appreciated the better inter-ventricular synchronization provided by the $RV_{apx} - LV_{bas}$ lead configuration when analyzing the histograms, which was also confirmed by a low TAT and recovery discrepancy between the SPH-based simulation and EAM data, as shown in Table 3. We can also see in Figure 6 the impact of changing the LV lead from the basal (or latest activated point) to the apex, increasing the inter-ventricular delay compared to the remaining configurations. Additionally, placing the RV lead in the RVOT was better than in the earliest activated point (i.e., lateral wall), as can be seen in Figure 6 (fourth and fifth row, respectively), the former having less IVD.

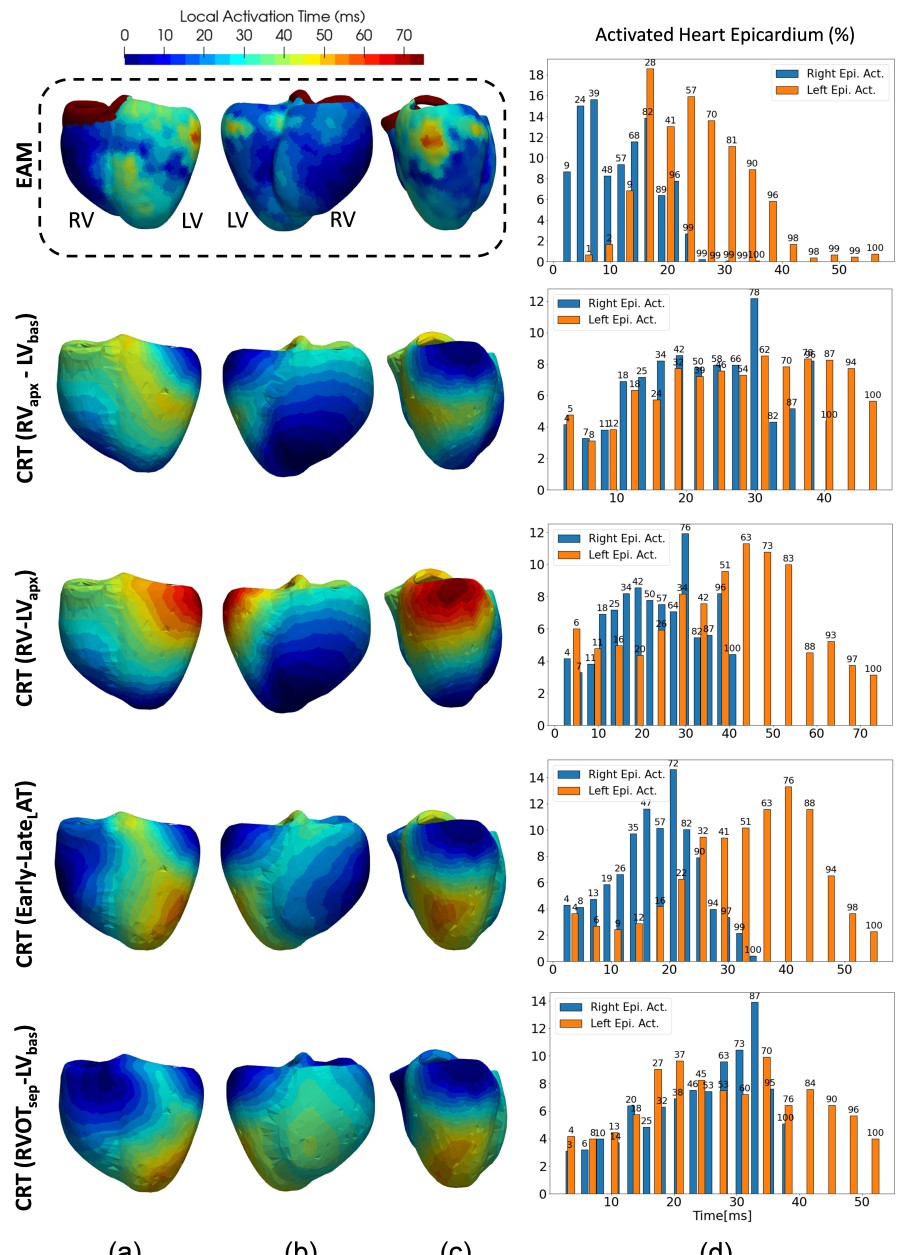

**Figure 6.** Local activation time maps ((**a**–**c**) showing anterior/posterior of biventricular epicardium and LV lateral wall epicardium, respectively) after cardiac resynchronization therapy (CRT) from the electro-anatomical (EAM) data and meshless simulations in the infarcted testing case, Pig 6. Histograms of the percentage of electrically activated heart tissue for the right and left epicardial layers are in the right column (**d**). From the second to the fifth row, different simulation results obtained with different CRT lead locations are displayed. $RV_{apx} - LV_{bas}$: leads on right ventricle (RV) apex and basal left ventricle (LV). $RV - LV_{apx}$: both leads are located in the biventricular apex. $Early - Late_L AT$: leads located at the the earliest and latest EAM ventricular activated points, respectively. $RVOT_{sep} - LV_{bas}$: leads in the septal RV outflow track and in the basal LV.

The non-infarcted cases of the testing database (Pig 4 and Pig 5) had an identical overall behavior with respect to optimal lead configuration. However, they presented large TAT errors between SPH-based simulations and EAM data (22.2 ms and 15 ms for Pig 4 and Pig 5, respectively), with CRT simulations providing lower TAT values and, consequently, larger recovery than EAM data (see Table 3). The difference between optimal SPH-based simulations and EAM measurements was due to a different lead location. Figure 7 shows

the histograms of the percentage of electrically activated heart tissue for both ventricles under LBBB and CRT conditions for the three analyzed testing cases, showing the large synchronization recovery achieved for all cases.

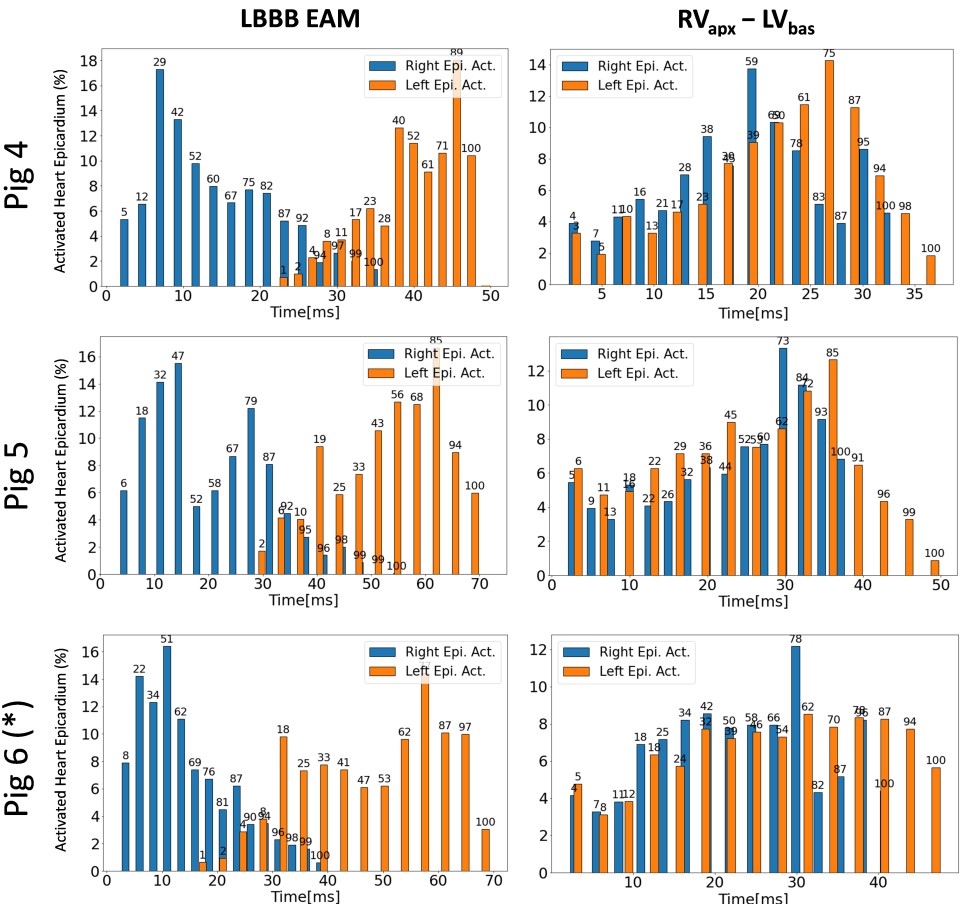

**Figure 7.** Percentage of electrically activated heart epicardium with a left bundle branch block (LBBB) and with the best cardiac resynchronization (CRT) lead configuration in the three testing cases. $RV_{apx} - LV_{bas}$: RV apex and LV basal stimulation regions. Epi: epicardium. act: Activation. (*) indicates an infarcted pig.

## 4. Discussion

Computational models of the heart can provide useful insight on the pathophysiological mechanisms and device options in CRT, contributing to reduce the high rate of non-responders. However, computational models need to be personalized and validated (after verification) with data coming from different sources, following standards such as the V&V40, to build the required credibility to be part of the device design and regulatory evaluation pipelines in silico trials [47]. Regrettably, it is not straightforward to acquire rich in vivo human data in clinical applications such as CRT. However, researchers (e.g., Rigol et al. [24,26]) have developed realistic experimental models, generating animal data that can be used to test and personalized the developed computational models of the heart.

The CRT-EPiggy19 challenge provided as open access multi-modal data of swine models under healthy, LBBB and CRT conditions, for model benchmark purposes. Three research teams [27–29] participated in the challenge, running different FEM-based electrophysiological models in patient-specific biventricular meshes that were provided by the organizers. In practice, mesh generation from patient-specific data of complex organs such as the heart often involves tedious manual interactions that hinder the application of computational models to large patient databases. Meshless models are an interesting alternative that have already been applied in cardiac electromechanics [33,36,38,39], but they

have not been benchmarked with FEM-based approaches in LBBB and CRT experimental data. Meshless methods are obviously independent of the generation of patient-specific cardiac meshes, thus solving one of the main bottleneck steps of mesh-based alternatives for translating computational models into clinical environment.

In this manuscript we present a meshless modeling pipeline, based on the SPH-based approach developed by Lluch et al. [38], where the most relevant parameters have been optimized to fit a subset of the CRT-EPiggy19 dataset. Basically, training data of three cases with LBBB were used to estimate the model parameters minimizing the differences between local activation times provided by meshless simulation results and EAM measurements, consequently predicting CRT electrical activation patterns with known lead location. Several metrics, proposed by Soto Iglesias et al. [25], were used, beyond the common global TAT parameter, to better quantify the local electrical heterogeneity in the ventricles. Although computational times could be further reduced, the meshless method could provide CRT predictions and lead configuration optimal strategies in around 20 min, once the LBBB pattern has been assimilated. Timings which are compatible with the clinical routine workflow. Moreover, meshless methods make the potential coupling with other physical models very easy, compared with FEM alternatives, with electromechanical models allowing large deformations without the risk of convergence issues due to mesh element quality degeneration.

The most relevant parameters related to the SPH-based model were the number of particles and the kernel size, which were set up to different values ($15 \times 10^3$ and 6.5–8.5 mm, respectively) than the original SPH formulation in [42] ($51 \times 10^3$ and 3 mm, respectively). The main reason was to decrease the computational cost for each simulation, without compromising result accuracy, so that the meshless method could be embedded into an parameter optimization framework. On the other hand, Mountris and Pueyo [39] employed a higher number of particles ($240 \times 10^3$) and a fixed neighbourhood size (150 particles) in their meshless model applied to CRT-EPiggy19 data, which would be prohibitive in our application due to the exponential growth in computational cost of the SPH-based solution.

### 4.1. Benchmark Analysis of Meshless and Finite-Element Method Solutions on Training Data

Despite the complex pipeline to process EAM data and the variability of the analyzed cases, the SPH-based model provided low LAT errors in LBBB ($6.75 \pm 1.59$ ms) and CRT ($10.38 \pm 3.80$ ms) cases. The meshless simulation results were generally similar to FEM-based ones from CRT-EPiggy19 participants (see Table 1 and Figure 5), when qualitatively analyzing the electrical activation patterns, and with the quantitative metrics (e.g., TAT, LAT, delays). However, some methodological differences were found that could explain small variations in the obtained results. For instance, the approach by Gomez and Sebastian [29], using a biophysical Ten Tusscher–Panfilov model, with a larger number of parameters and a more personalized Purkinje system differentiating between RV and LV, will certainly be more appropriate than simplified phenomenological Eikonal models in some cases. On the other hand, the low computational cost of Eikonal-based solutions allow running a lot of simulations and a larger exploration of the parameter space to match EAM data. Computational times for both meshless and FEM-based methods depend on the domain resolution (i.e., number of points/elements), the complexity of the electrophysiological model, and the number of parameters to estimate in the optimization procedure. Moreover, there is also a variety of IT resources involved. Independently of these factors, the main advantage of the meshless methods is the time saved to prepare the simulation domain compared to FEM alternative, which can be a matter of hours for complex geometries.

The key parameters related to the electrophysiological modeling for better fitting the EAM data were (1) the initial stimuli (number and position) of the electrical activation, (2) the modeling of the PK system, (3) and the regional conduction velocity distribution, which was optimized for each analyzed case. For instance, most participants [27,29] and ourselves adapted their modeling solutions to consider a possible retrograde activation of the PK system, via an extra-stimulus, to replicate the rapid activation from the apex to the LV base observed in the EAM data. Figure A3 in the Appendix shows how using only

one stimulus provided simulation results farther from the EAM data (error of 15.3 ms vs. 5.1 ms for two stimuli), demonstrating the dependence of the simulated activation patterns on the stimulation protocol. Potential causes for the PK retrograde activation might be the more transmural PK system in pigs compared to humans, which lead to incomplete LBBB such as in Pig 3 (anterior PK branch being functional while posterior branch being damaged, affecting the epicardial propagation). Modeling solutions with dedicated PK models such as in [29,39] could explain their better performance in these cases, justifying the use of more detailed and personalized PK system estimation algorithms [46,48].

The most important parameter to optimize in all electrophysiological modeling solutions to match EAM data were the regional distribution of conduction velocities, with computational costs directly linked to the chosen number of regions. We performed a sensitivity analysis that resulted in the use of 5 regions (RV/LV endocardium/epicardium, PK system), which avoided overfitting of LBBB-estimated results when applied to CRT cases (effect seen with a larger number of regions) and reasonable computational times (e.g., around 20 min per simulation). The CV distribution provided by the SPH-based model (see Table 2) are physiologically meaningful (e.g., PK being the fastest region, endocardial regions faster than epicardial ones, the scar having the lowest CV values), due to the imposed constraints in the optimization step. Cedilnik and Sermesant [28] used the same regions without PK in the only case they processed (Pig 3), however obtaining similar qualitative results in CRT simulations to Gomez and Sebastian [29] and ourselves. The regional strategies selected by Khamzin et al. [27] and Gomez and Sebastian [29] were the opposite, personalizing 30 and 34 (17 $AHA_{segments}$ division in both ventricles) regional parameters of conduction velocities, respectively, which gave them a lot of flexibility to match EAM data at the expense of risk of overfitting, as could be the reason of non-physiological CV distribution in some cases, compared with the SPH-based results (see ischemic case in Table 2, with conduction velocity slower in PK than in heart tissue).

Aiming at a perfect matching of simulation results to EAM data is not a simple task due to the variability of electrical patterns and the data uncertainty coming from the nature of EAM acquisitions and the post-processing (e.g., interpolation) required to create the biventricular meshes with local activation time maps. For instance, the sequential way (point-to-point) for acquiring the EAM data made the measurements dependent on the heart's anatomy and the number of EAM points, which was relatively low since an old system (CARTO XP) was used. Unexpected electrical activation patterns in the EAM of some cases could be explained by EAM interpolation effects. For instance, Pig 2 and Pig 4 non-infarcted cases had a significantly smaller amount of anatomical point-based acquisitions from CARTO XP in the LV anterior epicardial layer, leading to a 20 ms slower activation in the posterior vs. the anterior epicardial LV. We could not capture such heterogeneity in the SPH-based modeling pipeline since a single conduction velocity parameter was used for the entire LV epicardium, leading to the highest regional error in this area (see Tables A1 and A2 in the Appendix A).

Additionally, data uncertainty can lead to unrealistic and non-physiological parameters providing a better fitting between simulations and observations. For example, similar to Gomez and Sebastian [29], we needed high conduction velocities in areas near the scar in the infarcted cases (Pig 3 and Pig 6) to better fit EAM data with LBBB. Additionally, some participants included a second stimulus in the RV to better replicate the available electrophysiological measurements, which could correspond to the influence of the RV septomarginal trabecula but it could also be an interpolation artefact due to the sparsity of the EAM data. In the SPH-based modeling pipeline, we chose a constrained optimization algorithm to impose certain physiological requirements, at the expense of having less degrees of freedom, unlike approaches taken by other participants [27] that help them to achieve better fitting with EAM data (3–4.5% of LAT error in both LBBB and CRT training cases).

Another source of uncertainty is the position of the CRT leads, which justifies some differences between simulation results from all participants and EAM data. In Pig 1 and Pig 3 of the training dataset, the sub-optimal lead configuration was remarkable. In Pig 1, the apicality of the leads in both ventricles reduced the benefit of biventricular pacing, thus

being a CRT non-responder (low recovery metric) both in the meshless simulation and in real data. A different CRT configuration with the LV lead placed at the LBBB latest activation point improved the recovery for both meshless and FEM-based simulations. In Pig 3, the meshless and FEM-based simulations managed to improve the recovery percentage after a reduction of the TAT due to the estimated high conduction velocities. However, the basal configuration of the leads also both located in the epicardial layer of the LV for the high extension in the apical zone and transmurality of the scar, determined the ineffectiveness of biventricular resynchronisation therapy reflected in the delay metrics such as the IVD. An analogous behavior was found by Cedilnik and Sermesant [28] with a practically identical CRT prediction LAT error (6.5 ms and 6.6 ms for them and us, respectively) to SPH (6.6 ms).

The simulation protocol designed by CRT-EPiggy19 organizers asked to personalize model parameters with the LBBB data and use them to predict CRT measurements. However, some participants [27,29] applied correction strategies to better fit EAM data after CRT. Gomez and Sebastian [29] recalculated the conduction velocities for Pig 3, allowing a better match in the LV apical part than with the SPH-based model without corrections. Khamzin et al. [27] estimated a weight to adapt LBBB regional conduction velocities to CRT using Montecarlo random sampling and simulating 1000 different electrical activation patterns for each sample due to the low computational cost of their Eikonal-based model. Additionally, we did not use warming-up cardiac cycles to establish robust initial boundary conditions in the SPH-based model, while Gomez and Sebastian [29] had 10 cardiac cycles for stabilization purposes (taking 36 h), following the pipeline they previously optimized for arrhythmia simulation [49]. Although initial boundary conditions should not have a large influence for predicting activation maps, a rigorous study should be performed to confirm this assumption.

### 4.2. Validation of Meshless Method Results on Testing Data

Three testing cases of the CRT-EPiggy19 challenge were also processed with the SPH-based modeling pipeline. Lamentably, FEM-based simulation results were not available for benchmarking. As the CRT lead location was not provided in the testing cases, four different lead configurations based on literature [11,50–52] were evaluated. In the three analyzed testing cases, the optimal configuration was with a RV apical lead and the LV one placed at the epicardial lateral wall ($RV_{apx} - LV_{bas}$ in Figures 6, A4 and A5). The $RV_{apx} - LV_{bas}$ lead configuration not only provided better recovery percentages, but also had a smaller LAT error with CRT EAM data (see Table A2 in the Appendix A). This is in agreement with multiple clinical studies and guidelines [11,53], although different alternatives (e.g., different RV location [20]) are still being proposed. For instance, some studies suggest that RVOT pacing may be more beneficial than standard one, specifically in cases with a decreased left ventricular ejection fraction [11,50]. In our study, RVOT pacing was the second best lead configuration, but still with slightly worse overall efficiency compared to $RV_{apx} - LV_{bas}$. The worst scenario was when both leads were in apical locations, as in the case of Pig 1, where the benefits of bi-ventricular pacing are reduced to only one lead due to an overlap of the electrical breakthrough waves.

### 4.3. Limitations and Future Work

The presented study has several limitations at different levels. First, the available data from the CRT-EPiggy19 challenge ertr useful to identify and better understand key aspects of different CRT models. However, several factors associated with EAM acquisition and processing induced a non-negligible data uncertainty that can limit the conclusions from the study. As well, hemodynamic descriptors, e.g., based on Doppler-derived measurements, were not available from the experimental study in Rigol et al. [24], preventing the optimization of important CRT parameters such as the AV delay, which has been found a potential non-responder factor [54]. Moreover, even in the case of better animal experiments, models should also be tested on in vivo human data to investigate its added value in the CRT clinical pipeline. Furthermore, the processing and modeling of each case, including a large number of simulations for parameter optimization, is very time consuming. The conse-

quence is that only a few cases could be processed in our study and by the CRT-EPiggy19 participants, limiting the impact and generalizability of the benchmark analysis. A more comprehensive comparison with other FEM-based and meshless models in common data would be beneficial.

The proposed SPH-based modeling pipeline provided simulation results comparable to the state-of-the-art alternatives, but several improvements could be incorporated. Firstly, the inclusion of the anisotropic ratio and the myocardial layer for each ventricle in the optimization pipeline could give more degrees of freedom to match EAM data. Additionally, the parameter optimization schemes used by all participants of the CRT-Epiggy19 challenge were not taking advantage of recent technological advances such as the use of deep learning algorithms [55,56], variational approaches [57], reduced-order models [58,59] or GPU-based architectures [60], which allows for the exploration of a larger space of parameter solutions at reduced computational times. Moreover, cardiac multi-physical models should provide more realistic simulations, allowing for the inclusion of hemodynamic factors and improving the adjustment of CRT configuration through flow ratios [61], perfusion models [17], lumped models of the whole cardiovascular circulation [18] or with a complete torso [20].

## 5. Conclusions

A meshless modeling pipeline to simulate cardiac electrical patterns in CRT was compared to FEM-based alternatives, providing equivalent results on fitting experimental data available from the CRT-EPiggy19 challenge. The main advantage of the meshless model is the independence from the usually arduous patient-specific meshing process, one of the most important bottlenecks of translating computational models into a clinical environment. However, the most relevant aspect for accurate CRT predictions was the chosen parameter personalization strategy rather than the geometrical discretization. In particular, the regional conduction velocity distribution was key, requiring at least five different regions and ideally including a PK label. A larger number of regions was associated with better data fitting but higher computational costs and more risk of overfitting. Additionally, the optimal CRT configuration was found with apical RV and basal LV leads, as reported in the literature. Despite the uniqueness of the CRT-EPiggy19 challenge dataset, data uncertainty was high in some cases due to challenging EAM acquisition and processing, which could lead to the estimation of non-physiological parameters and the requirement of prior constraints in the optimization algorithm. Nevertheless, having several teams of modeling researchers working on the same data have been beneficial for each challenge participant, jointly improving the different modeling solutions.

**Author Contributions:** Conceptualization, C.A., J.F.G., N.C., K.A.M., T.M., S.K., A.D., O.S., E.P., M.S., R.S., H.G.M. and O.C.; Data curation, T.M.; Formal analysis, C.A., J.F.G., N.C., K.A.M., S.K., A.D., O.S., E.P., M.S. and R.S.; Investigation, C.A., J.F.G., N.C., K.A.M., S.K., A.D., O.S., E.P., M.S., R.S. and O.C.; Methodology, C.A., J.F.G., N.C., K.A.M., S.K., A.D., O.S., E.P., M.S. and R.S.; Project administration, O.C.; Software, È.L. and H.G.M.; Supervision, O.S., E.P., M.S., R.S. and O.C.; Writing—original draft, C.A., J.F.G., N.C., K.A.M., S.K., A.D., O.S., E.P., M.S., R.S. and O.C.; Writing—review & editing, C.A. and O.C. All authors have read and agreed to the published version of the manuscript.

**Funding:** This work was supported in part by the Spanish Ministry of Science and Innovation (TIN2011-28067, REDINSCOR RD06/003/008, PID2019-105674RB-I00), the Spanish Industrial and Technological Development Center (cvREMOD-CEN-20091044), the Spanish Ministry of Economy and Competitiveness under the Maria de Maeztu Units of Excellence Programme (MDM-2015-0502), the Government of Aragón (Spain) (LMP94_21), the Directorate General of Science Policy of the Generalitat Valenciana (Spain) (PROMETEU 2016/088), the Seventh Framework Programme (FP7/2007-2013) for research, technological and demonstration under grant agreement VP2HF (no. 611823), by the RSF grant no. 19-14-00134, and by European Regional Development Fund (ERDF) DPI2016-75799-R (AEI/ERDF, EU).

**Institutional Review Board Statement:** The present manuscript was based on a subset of the data from the experimental study conducted by Rigol et al. [24] at Hospital Clinic de Barcelona in which animal handling was approved by the Institutional Review Board and Ethics Committee (approval reference number: DMAH 5648). Additional information can be found at: Rigol M, Solanes N, Fernandez-Armenta J, Silva E, Doltra A, Duchateau N, Barcelo A, Gabrielli L, Bijnens B, Berruezo A, Brugada J, Sitges M. Development of a swine model of left bundle branch block for experimental studies of cardiac resynchronization therapy. J Cardiovasc Transl Res. 2013 Aug;6(4):616-22. doi: 10.1007/s12265-013-9464-1.

**Conflicts of Interest:** The authors declare no conflict of interest.

**Appendix A**

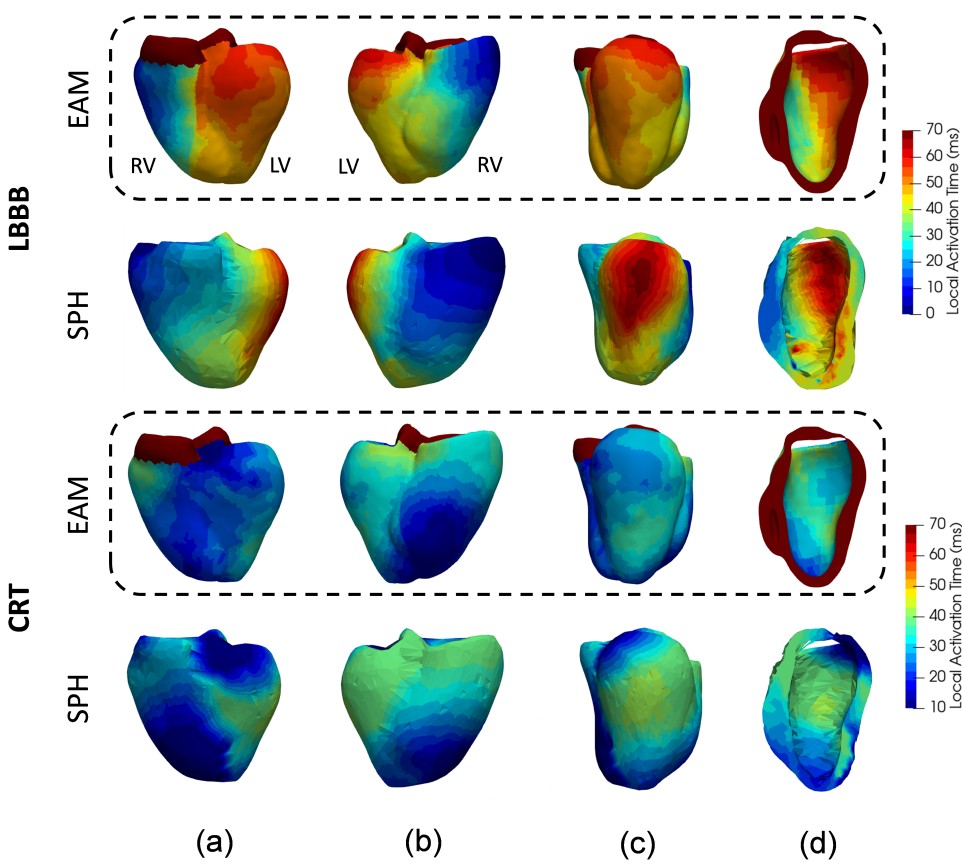

**Figure A1.** Local activation time maps for a non-infarcted case, Pig 2, of the training database in left bundle branch block (LBBB) and cardiac resynchronization therapy (CRT) conditions, provided by the electroanatomical measurements (EAM) and a meshless (SPH) model. (**a**,**b**) correspond to anterior and posterior biventricular epicardial visualizations. (**c**,**d**) show epicardial and endocardial view of the left ventricle (LV) lateral wall, respectively. RV: right ventricle.

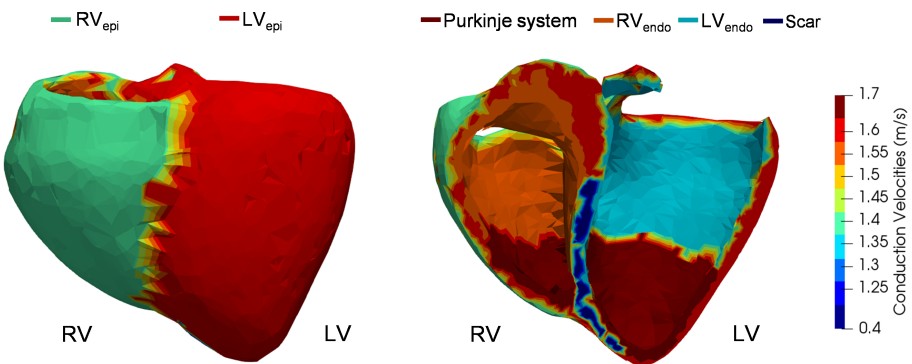

**Figure A2.** Conduction velocity map for one of the ischemic cases analyzed in our study.

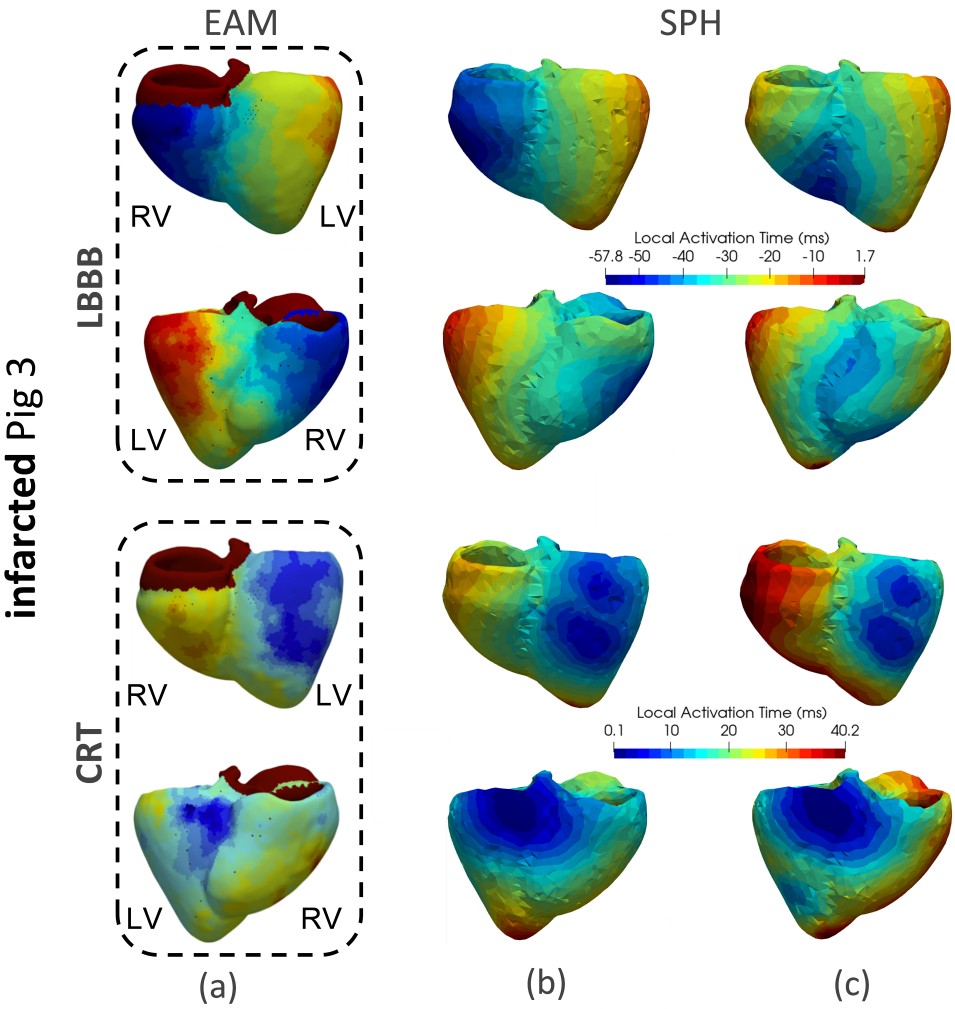

**Figure A3.** Local activation time maps for the infarcted case, Pig 3, of the training database in left bundle branch block (LBBB) and cardiac resynchronization therapy (CRT) conditions, provided by the (**a**) electroanatomical measurements (EAM) and the (**b**,**c**) meshless (SPH) model. From top to bottom in each condition, the anterior and posterior biventricular epicardial visualizations are shown, respectively. The electrical activation patterns acquired by maintaining the strategy of two initial stimuli (Right Ventricle (RV) and septal) are represented in (**b**) and disregarding only the initial RV stimulus in (**c**). LV: left ventricle.

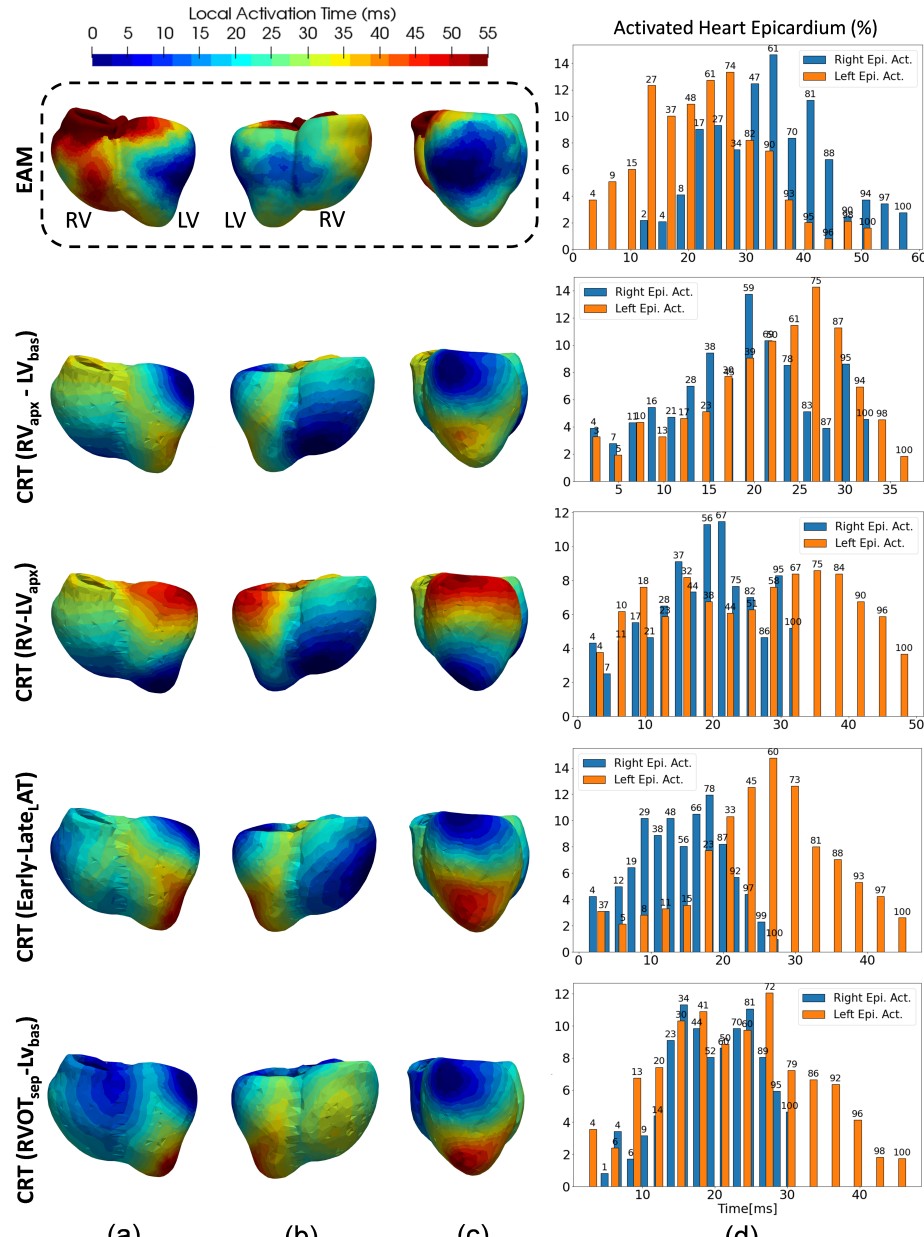

**Figure A4.** Local activation time maps ((**a**–**c**) showing anterior/posterior of biventricular epicardium and LV lateral wall epicardium, respectively) after cardiac resynchronization therapy (CRT) from the electro-anatomical (EAM) data and meshless simulations in the non-infarcted testing case, Pig 4. Histograms of the percentage of electrically activated heart tissue for the right and left epicardial layers are in the right column (**d**). From the second to the fifth row, different simulation results obtained with different CRT lead locations are displayed. $RV_{apx} - LV_{bas}$: leads on right ventricle (RV) apex and basal left ventricle (LV). $RV - LV_{apx}$: both leads are located in the biventricular apex. $Early - Late_L AT$: leads located at the the earliest and latest EAM ventricular activated points, respectively. $RVOT_{sep} - LV_{bas}$: leads in the septal RV outflow track and in the basal LV.

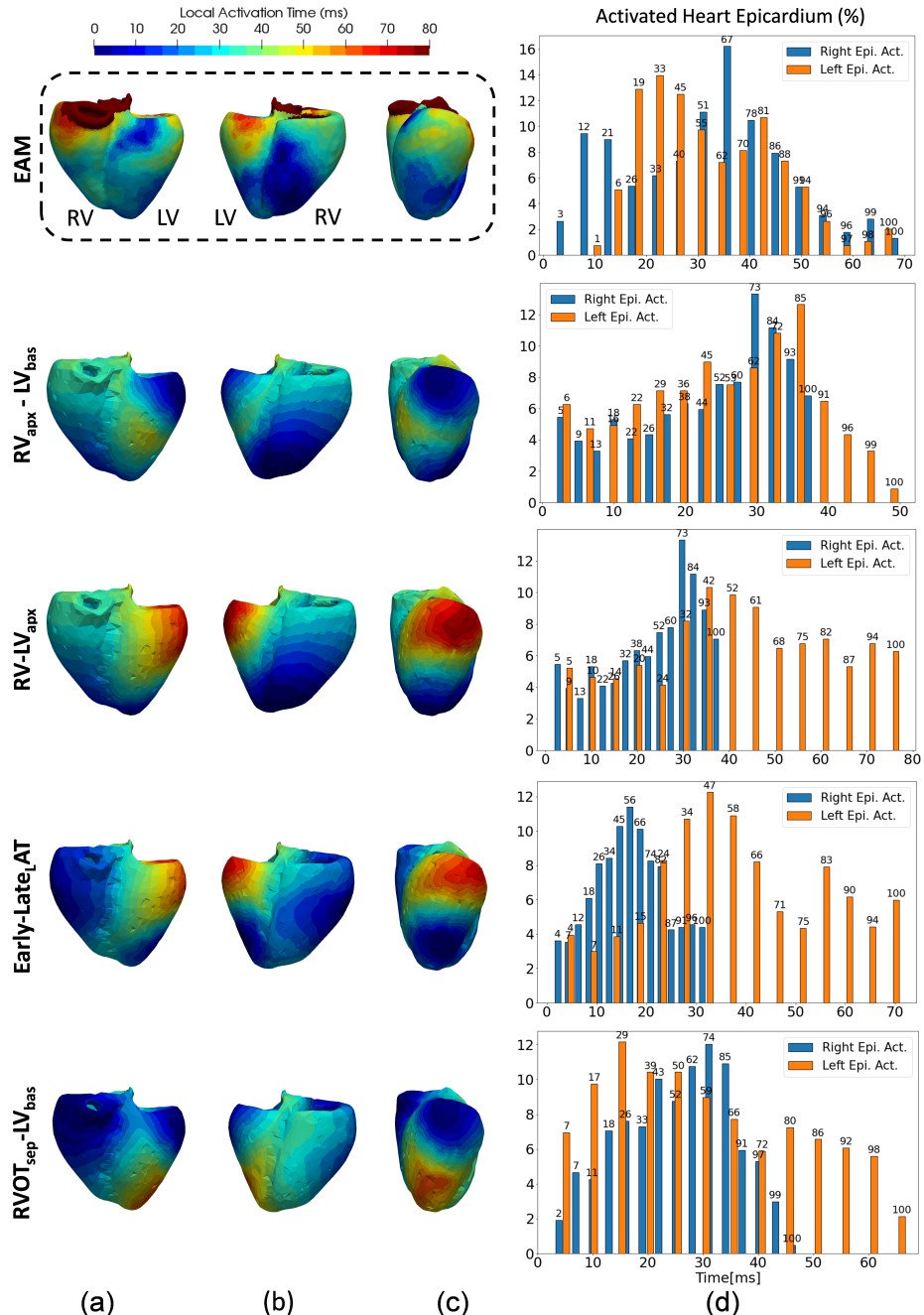

**Figure A5.** Local activation time maps ((**a**–**c**) showing anterior/posterior of biventricular epicardium and LV lateral wall epicardium, respectively) after cardiac resynchronization therapy (CRT) from the electro-anatomical (EAM) data and meshless simulations in the non-infarcted testing case, Pig 5. Histograms of the percentage of electrically activated heart tissue for the right and left epicardial layers are in the right column (**d**). From the second to the fifth row, different simulation results obtained with different CRT lead locations are displayed. $RV_{apx} - LV_{bas}$: leads on right ventricle (RV) apex and basal left ventricle (LV). $RV - LV_{apx}$: both leads are located in the biventricular apex. $Early - Late_L AT$: leads located at the the earliest and latest EAM ventricular activated points, respectively. $RVOT_{sep} - LV_{bas}$: leads in the septal RV outflow track and in the basal LV.

**Table A1.** Quantitative measures characterising the regional local activation time error in the electrical activation maps for the training cases from meshless simulations. LBBB: left bundle branch block. CRT: cardiac resynchronization therapy. RMSE: root mean square error. LV: left ventricle. RV: right ventricle. Epi: epicardium. Endo:endocardium. (*) indicates an infarcted pig.

| RMSE (ms) | Pig 1 | | Pig 2 | | Pig 3 (*) | |
|---|---|---|---|---|---|---|
| | LBBB | CRT | LBBB | CRT | LBBB | CRT |
| **RV Epi** | 5.6 | 8.73 | 7.95 | 7.04 | 5.6 | 4.3 |
| **LV Epi** | 5.9 | 7.37 | 10.17 | 8.53 | 4 | 6.2 |
| **LV Endo** | 9.1 | 7.45 | 9.76 | 7.48 | 6.4 | 9.7 |

**Table A2.** Quantitative measures characterising the regional local activation time error in the electrical activation maps for the testing cases from meshless simulations. LBBB: left bundle branch block. CRT: cardiac resynchronization therapy. RMSE: root mean square error. Epi: epicardium. Endo:endocardium. $RV_{apx} - LV_{bas}$: right ventricle apex and basal region of the left ventricle. $RV - LV_{apx}$: RV apex and LV apex. $Early - Late_L AT$: earliest and latest EAM activation points. $RVOT_{sep} - LV_{bas}$: RV outflow track septal and LV basal region. (*) indicates an infarcted pig.

| | RMSE (ms) | RV Epi | LV Epi | LV Endo |
|---|---|---|---|---|
| **Pig 4** | LBBB | 3.9 | 5.47 | 6.42 |
| | CRT ($RV_{apx} - LV_{bas}$) | 17.14 | 11.6 | 10.71 |
| | CRT ($RV - LV_{apx}$) | 17.23 | 14.56 | 13.16 |
| | CRT ($Early - Late_L AT$) | 20.67 | 13.4 | 11.86 |
| | CRT ($RVOT_{sep} - LV_{bas}$) | 15.46 | 13.17 | 12.33 |
| **Pig 5** | LBBB | 4.92 | 7.68 | 13.4 |
| | CRT ($RV_{apx} - LV_{bas}$) | 10.98 | 16.26 | 17.85 |
| | CRT ($RV - LV_{apx}$) | 10.98 | 14.74 | 19.27 |
| | CRT ($Early - Late_L AT$) | 20.87 | 10.82 | 14.73 |
| | CRT ($RVOT_{sep} - LV_{bas}$) | 19.82 | 21.88 | 20.8 |
| **Pig 6 (*)** | LBBB | 5.97 | 6.11 | 5.7 |
| | CRT ($RV_{apx} - LV_{bas}$) | 11.94 | 12.21 | 7.98 |
| | CRT ($RV - LV_{apx}$) | 12.03 | 18.95 | 14.01 |
| | CRT ($Early - Late_L AT$) | 5.91 | 14.48 | 12.7 |
| | CRT ($RVOT_{sep} - LV_{bas}$) | 13.78 | 12.85 | 11.93 |

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
