# Peer review of "Meshless Electrophysiological Modeling of Cardiac Resynchronization Therapy—Benchmark Analysis with Finite-Element Methods in Experimental Data"

_applsci, doi:10.3390/app12136438_

Round 1

Reviewer 1 Report

I read with great interest this manuscript on modelling for cardiac resynchronisation therapy. I am not an expert on the tecniques that were employed but have some comments on clinical context:

The references are somewhat outdated. The guidelines from 2016 for heart failure have been updated. Te introduction needs some improvement.

Last but not least it is difficult to understand what the implications are and this section needs to be improved. CRT non-response is multifaceted and depending on several factors including optimal programming. AV synchrony is not included in the model and is an important part of resnchronisation.

Author Response

We are grateful to Reviewer 1 for expressing great interest in reading our manuscript. Please, find below the attachment in PDF with the response to each of the points raised by the reviewer. We found them very useful, and we are confident that the reviewer’s comments have helped us to improve the manuscript

Reviewer 2 Report

The manuscript Meshless electrophysiological modelling of cardiac resynchronization therapy - Benchmark analysis with finite-element methods in experimental data presents a meshless-based modeling method for the prediction of electrical patterns in cardiac resynchronization therapy. The manuscript is written well, and the results obtained during this study are presented clearly. Still, I have some remarks that might help to improve this manuscript:

  1. Some brief information on image segmentation and geometry reconstruction would be useful.
  2. Validation of the results is not so clear, some comparison with other authors was described in Results section, and some in Discussion. I would suggest including a separate subsection for validation of the results in the Discussion section.
  3. More details regarding the advantages of meshless models vs mesh-based models needs to be provided. Please include the comparison of required computational time for each simulation type, if available.
  4. Data labels in figures 5d, 6, A2, and A3 are poorly visible.
  5. Please consider checking language and style. For example, there are some spelling mistakes in the abstract:
    • “Meshless models can be a valid alternative due to its mesh quality independence”.
    • “We present here a benchmark analysis the a meshless-based method with finite-element methods”.
    • Please check this sentence, it doesn’t sound right: “The simulation results obtained with the meshless model were equivalent to FEM methods, being more critical the parameter personalization strategy <…> rather than the geometrical domain discretization”.
  6. As stated in the Limitations subsection, the proposed meshless-based modeling method is very time-consuming. What possible applications do authors see for such method at the moment?

Author Response

We are grateful to Reviewer 2 for a positive assessment of the content of the manuscript.
Please, find below the attachment PDF with the response to each of the points raised by the reviewer. We found them very useful, and we are confident that the reviewer’s comments have helped us to improve the manuscript.

Reviewer 3 Report

The manuscript proposes an interesting computational analysis comparing meshless methods versus classical finite element approaches applied to cardiac electrophysiology. I appreciate, in general, the overall spirit of the work. However, as detailed below, I have two major concerns preventing me from accepting the manuscript in its present form. I request the authors to perform a throughout revision of the manuscript, improving both descriptive and methodological sections.

Though references to review studies, the introduction is missing a critical part of the literature closely related to the present work. (1) Approaches based on high-speed GPU implementations, nonlinear diffusion, fractional Laplacian, stochastic gap junction models, etc., must be linked appropriately to the present study and its future directions. (2) Similarly, recent data estimation and assimilation procedures, reduced-order modeling approaches, prediction of chaotic behavior, neural networks, etc., must be mentioned appropriately. (3) Studies related to infarcted hearts and fibrillation sustainability.

A partial list of references is provided below.

Section 2.2 requires a better description of the adopted meshless method. The reader has no idea about the Gaussian Kernel usage and how it can be implemented in the case of anisotropy. Both a figure and the associated equations shall be briefly mentioned. Myocyte orientation is obscure to this reviewer, though it is a critical point for CV estimation.

CV estimation is not well-described. What is the direction of propagation? How many simulations were performed? Here, I have another major problem with the physical time simulated. The authors claim that they use a time resolution of 10^-4 (how is it fixed?) for a total time of 0.15s. This is methodologically wrong for two reasons: 
1) it is well known that cardiac electrophysiological simulations are affected by initial and boundary conditions. Accordingly, the usual strategy is to run a first excitation and disregard it, then apply a second stimulus that is used for the analysis;
2) the propagation of the action potential requires several hundreds of ms to reach the whole ventricles. Then, in the presence of anisotropy, it must be tuned according to the adopted direction and anisotropy ratio. 
Therefore, this preliminary test must be rebuilt in a more consistent way.

The authors do not provide any notation or nomenclature for the mathematical modeling part. In particular, they do not introduce vectors, tensors, and dyadic products properly (they should be boldface). Furthermore, they use a C symbol to describe the diffusion tensor without considering the classical use of D. Please correct this section in a more intelligible way for expert readers and do not spell C as a connectivity tensor. It is a diffusivity tensor based on cable equation derivation. In fact, c is not the sole term responsible for propagation speed (better than driving). It is known that both c and nonlinear reaction functions are responsible for the CV value.

The description of the Purkinje fiber structure is very obscure. It is mandatory to introduce a figure and describe how the myocardium and PK are modeled. Retrograde activation is controversial due to the localized junctions among Purkinje and myocardium. Why these numbers, 500 300? The reader has no justification for that. Again, it is mandatory to have a first stimulus disregarded. Why two locations with the same stimulus timing? This is not correct to establish a benchmark.
Besides, the figure is complex to understand because of the multiple surfaces plot. In fact, the reader has no information about the local activation time.
Lines 197-206: This is not clear, why “independent”? Maybe the authors mean “heterogeneous”. This is, however, a major limitation, see references. The authors are also wrong in reference to [23], which considers the Eikonal model. It is, on the contrary, well known that the local multiscale nature of the cell-cell coupling has an enormous effect on the macroscopic CV (see, Barone, Treml, Hurtado, etc).
Also, the sentence “slow electrical …” is not correct. The anisotropic nature of the cardiac tissue makes CV higher, not slower. Then, the study requires 21 regions, which represent highly heterogeneous materials. Does the problem come from the local discontinuities of the c values?
Actually, there is no such a map.
More advanced techniques for data estimation are not acknowledged (see references).

Line 245: "After sensitivity analysis".
What does it mean? What are the ranges of parameters? How many simulations? What is the computational cost?
There is no figure for CV distribution.

What is the color code in Figure 3?

The difference in RMSE between 21 and 5 regions does not justify the usage of 21 regions. Besides, with CRT the situation is also worse. In the end, the choice of 5 regions is not based on any optimality criterium and, in particular, not on CV. In fact, I request the authors to run the optimization study for the second physiological activation and longer times. The presence of anisotropies requires this. In the end, are we sure there is an advantage wrt FEM?

Problems related to the apex are known because of a complex anisotropy distribution.

Correct the following sentences:

Line 8: We present here a benchmark analysis the a meshless-based
Line 194: the two stimulus
Fig.2: pattern to activate the whole biventricular geometries to activate,
Line 230: In the testing cases of testing
Line 455: into an parameter

Partial list of references:

Shahi et al. https://www.sciencedirect.com/science/article/pii/S2666827022000275
Kaboudian et al. https://ieeexplore.ieee.org/abstract/document/9662759
Hurtado et al. https://www.sciencedirect.com/science/article/abs/pii/S0045782515003679
Cherubini et al. https://www.sciencedirect.com/science/article/pii/S0022519317303417
Barone et al. https://www.sciencedirect.com/science/article/abs/pii/S0045782519304918
Barone et al. https://www.sciencedirect.com/science/article/pii/S0021999120305842
Loppini et al. https://www.frontiersin.org/articles/10.3389/fphys.2018.01714/full
Cusimano et al. https://www.sciencedirect.com/science/article/abs/pii/S100757041930471X
Cusimano et al. https://aip.scitation.org/doi/abs/10.1063/5.0050897
Ramirez et al. https://www.nature.com/articles/s41598-020-69900-4
Treml et al. https://www.mdpi.com/2227-7390/9/2/164

Author Response

We are grateful to Reviewer 3 for a positive assessment of the content of the manuscript.
Please, find below the response in PDF format to each of the points raised by the reviewer. We found them very useful, and we are confident that the reviewer’s comments have helped us to improve the manuscript

Round 2

Reviewer 2 Report

I feel that the revised manuscript was well improved and can be accepted for publication.

Author Response

We are grateful to reviewer 2 for his assessment after re-checking the manuscript and proposing it for publication in the journal.

Sincerely,

Authors

Reviewer 3 Report

The authors notably improved the manuscript concerning the introduction, description of methods, results, and discussion.

I acknowledge the specific improvement done in terms of PK modeling, numerical tuning, and material properties description. I find the work fully reproducible in its present form and self-consistent, thus providing all the necessary information.

Regarding figure 2 provided in the cover letter, I think it is actually very informative for the reader and makes the work even more reproducible. I thus suggest introducing this figure in the manuscript. If the authors think it appropriate, this figure can go in an Appendix.

I appreciate the effort made by the authors in replying to my previous concerns, and I acknowledge the authors' experience in cardiac modeling, as well. However, I am still convinced that the obtained activation timings are affected by the stimulation protocol. I may suggest running only a single representative example to verify that the assumption of using only one stimulation returns the same quantity in activation times. This may give much more robust support to the adopted methodology.

Finally, the authors should take care of some typos and organization error:

line 296: please complete the sentence

line 354: table is wrong

line 355: missing reference link

line 374: missing reference link

line 375: space for the paragraph

line 389: missing reference link

line 399: missing reference link

line 490: please complete the sentence

REFERENCES: please, amend all the Journal titles. They need capital letters.

Author Response

We would like to thank reviewer 3 again for the proposed suggestions. We believe they have improved the robustness of the manuscript. Please, see the attached PDF where all comments made by the reviewer are addressed point by point,

Sincerely,
Authors
